# Text Sentiment Classification Based on BERT Embedding and Sliced Multi-Head Self-Attention Bi-GRU

**DOI:** 10.3390/s23031481

**Published:** 2023-01-28

**Authors:** Xiangsen Zhang, Zhongqiang Wu, Ke Liu, Zengshun Zhao, Jinhao Wang, Chengqin Wu

**Affiliations:** College of Electronic and Information Engineering, Shandong University of Science and Technology, Qingdao 266590, China

**Keywords:** sentiment classification, BERT word vector, bidirectional slice-gated recurrent unit, multi-head self-attention mechanism

## Abstract

In the task of text sentiment analysis, the main problem that we face is that the traditional word vectors represent lack of polysemy, the Recurrent Neural Network cannot be trained in parallel, and the classification accuracy is not high. We propose a sentiment classification model based on the proposed Sliced Bidirectional Gated Recurrent Unit (Sliced Bi-GRU), Multi-head Self-Attention mechanism, and Bidirectional Encoder Representations from Transformers embedding. First, the word vector representation obtained by the BERT pre-trained language model is used as the embedding layer of the neural network. Then the input sequence is sliced into subsequences of equal length. And the Bi-sequence Gated Recurrent Unit is applied to extract the subsequent feature information. The relationship between words is learned sequentially via the Multi-head Self-attention mechanism. Finally, the emotional tendency of the text is output by the Softmax function. Experiments show that the classification accuracy of this model on the Yelp 2015 dataset and the Amazon dataset is 74.37% and 62.57%, respectively. And the training speed of the model is better than most existing models, which verifies the effectiveness of the model.

## 1. Introduction

Sentiment analysis, known as opinion mining, is a research hotspot in the field of Natural Language Processing (NLP). The concept was originally proposed by Nasukawa et al. [1], aiming to analyze, identify, and mine people’s emotional tendencies. Sentiment analysis has a wide range of applications, including public opinion monitoring [2], product analysis [3], topic inference [4], etc. In terms of public opinion monitoring, relevant government departments can keep abreast of the public opinion trends of netizens, monitor the direction of public opinion propagation, and consequently adjust policies reasonably to avoid deterioration of the situation. Regarding the product recommendation, the sellers can survey satisfaction on time based on customer reviews. Thereby the product quality can be improved. In the aspect of topic inference, the computer can infer emotional changes in the article based on the previous content, and analyze the correlation between contexts, so as to reasonably infer the emotional tendency of the article.

Affective computation includes sentiment and other emotions expressed in a document, which can be captured and evaluated by opinion-mining algorithms. Typically, such algorithms are based on deep learning, lexical methods, or a combination of both [5]. Nowadays, the technology of sentiment analysis has developed rapidly. By using deep learning technologies to automatically capture the semantic features of the text, there is no need to manually establish the feature space. The pre-trained word representation is an important part of NLP tasks. Traditional word vector models, such as Word2Vec [6], Glove [7], etc., are all static word vectors, which will cause the same word vector representation of certain words in different contexts. Therefore, the meaning of the words cannot be accurately represented in the current context.

At present, the mainstream model for sentiment analysis tasks is a combination of Recurrent Neural Network (RNN) (including LSTM) and attention mechanism. However, the standard RNN structure has the shortcomings, such as time step dependence during the training process and lack of parallel training ability, leading to slower training speed. Although the Convolutional Neural Network (CNN) can capture the local feature information and implement parallel operations, its shortcomings are also obvious. It is unable to fully obtain the timing information of the sequence, so it cannot learn the context information of the text. As far as the attention mechanism is concerned, the traditional attention mechanism [8] obtains the weight of features through dot multiplication or cosine similarity. But it can only calculate the weight values of different words without inferring the relationship between the words in the sequence.

To overcome the drawbacks of the above models, we propose a new model for text sentiment classification.

Firstly, we propose a Sliced Bidirectional Gated Recurrent Unit (Sliced Bi-GRU) as a straightforward enhanced version of Sliced Recurrent Neural Networks(SRNN) [9]. It is very easy to understand that Sliced Bi-GRU integrates the merits of Slice operation, Bi-direction, and GRU over the Classical RNN.

Then, In order to further improve the performance of the model, adopting the Bidirectional Encoder Representations from Transformers (BERT) word vector representation as the input of the model embedding layer, we combine our Sliced Bi-GRU structure and the Multi-head Self-Attention (MSA) mechanism [10] to propose a novel model.

By adding the MSA mechanism to each sub-sequence, the associated information can be captured between the words. The reason for the extension to Multi-head attention is that the parameters of each head are different, and various sub-spaces information can be learned during the calculation process. Another advantage of Multi-head attention is that it can prevent the model from overfitting. This article conducts experiments on the Yelp 2015 review dataset and Amazon Product review dataset. Compared with mainstream sentiment classification models, the proposed model has achieved better results in classification accuracy and training speed, which proves the effectiveness of the model.


**Main Contribution**


It is the first time that we propose the concept of a Bidirectional Slice Gated Recurrent Unit in this article. The slicing network can speed up the training of the model. Meanwhile, the model can reach the balance of performance and efficiency.It is the first time that we combine the multi-head self-attention mechanism with the sliced bi-directional Gated Recurrent Unit, which can learn hidden information in different subspaces.We take advantage of BERT nature to perform pre-training. The Sliced Bi-GRU can model each part in various individual sub-space. The MSA helps the model to increase semantic relevance issues. As a result, the performance of the proposed method keeps better by combination of BERT, Sliced Bi-GRU, and MSA.

## 2. Related Works

In recent years, the application of deep learning-related technologies in text sentiment classification has developed rapidly. Kim et al. [11] first applied CNN to NLP tasks and proposed a new algorithm—TextCNN. They input the word vectors into the CNN network to extract semantic information to implement the task of text classification. Ouyang et al. [12] proposed a combination of CNN network and Word2Vec word vectors. In order to prevent the model from overfitting, the dropout technology was complemented to improve the generalization performance of the model. Huang et al. [13] established a tree-like LSTM network structure, which can be used to model sentence-level text data to improve the classification ability of the model.

Bradbury et al. [14] combined CNN with RNN networks to construct a new structure—Quasi Recurrent Neural Networks (QRNNs), whose training speed is 16 times faster than the simple RNN structure. Xue et al. [15] incorporated CNN with the gate mechanism and took the Tanh-Relu activation function as the threshold unit to capture the corresponding emotional features. Yu et al. [9] improved the structure of the RNN network and proposed a Sliced Recurrent Neural Network (SRNN). This network structure can divide the original sequence into smaller sequences, thereby realizing a parallel RNN network and accelerating the training speed while maintaining good classification performance.

The nearly most powerful NLP model released by the Google team—BERT [16], broke the best record of 11 NLP tasks in one fell swoop and became the most successful model in the NLP field. The innovation of BERT lies in the use of the Transformer structure and the mask processing in the pre-trained language model. This method can fully describe the relationship between character-level [17], word-level, and sentence-level. Therefore, the generalization ability of the pre-trained language model can be further improved.

The application of the attention mechanism [18] in text sentiment analysis tasks has attracted more and more researchers’ attention. Zhang et al. [19] utilized CNN to extract features and then calculated the attention weight of extracted features. Mishev et al. [20,21,22] adopted Bi-sequence Gated Recurrent Unit (Bi-GRU) to extract the emotional features in the text and evaluated the model via the attention mechanism. Yang et al. [23] proposed a Hierarchical Attention Network (HAN), which applies different attention mechanisms to words and sentences in the text from two dimensions, respectively. While maintaining the integrity of the document structure, it can also extract different levels of emotional feature information.

In 2017, Google proposed a Self-attention mechanism, which achieved the best performance in machine translation at that time. Chen et al. [24] combine the RNN and the attention mechanism to obtain richer text information to improve the classification accuracy of the model. Cambria E et al. [25] proposed symbolic, and sub-symbol AI tools, which integrate top-down and bottom-up learning and integrate their logical reasoning into the deep learning framework to build a commonsense knowledge base for sentiment analysis. Ambartsoumian et al. [26] applied the Self-attention mechanism to the sentiment analysis task. They started with three indicators of classification accuracy, training speed, and memory consumption and conducted comparative experiments on six datasets. The evaluation model verifies the advantages of the Self-attention mechanism. Shen et al. [27] proposed a Directed Self-Attention Network (DiSAN) model in which the attention between elements from the input sequence(s) is directional and multi-dimensional. The model compresses the sequence into a multi-dimensional vector representation. The structure is composed of directed Self-attention blocks with time sequence coding, which has high test accuracy in all sentence coding methods. M. S. Akhtar et al. [28] propose a stacked ensemble method for predicting the degree of intensity for emotion and sentiment by combining the outputs obtained from several deep learning and classical feature-based models using a multi-layer perceptron network. E. Cambria et al. [29] proposed an Attention-based Bidirectional CNN-RNN Deep Model (ABCDM)to solve the equally important problem of different functions.

## 3. The Proposed Approach

As described previously, we adopt the BERT word vector representation as the pre-trained input of the model and then combine the Bi-SGRU and MSA mechanism to address a text sentiment classification model. Firstly, the dynamic word vector representation is obtained by the BERT pre-training language model, which can represent text information more accurately. At the same time, we slice the text sequence into subsequences of equal length and extract the features for each sub-sequence independently, which realizes the parallelization of the neural network and accelerates the training speed.

### 3.1. BERT Pre-Trained Language Model

In recent years, utilizing the pre-trained deep neural network as the language model and then fine-tuning [30] the downstream tasks has been the focus of research. These word vectors need to accurately represent the meaning of words and the characteristics of sentences. The BERT model solves these problems well. It adopts a bidirectional Transformer as the framework for training the model and uses two new strategies when training the model.

The “mask” language technics is applied for word-level representation. Specifically, the training data generator randomly selects 15% of the token word positions in each sentence to be randomly masked. And the model predicts the masked words based on their context information. If the i-th token is selected, we will replace the i-th token with (1) a [MASK] token with an 80% probability, (2) a random token with a 10% probability, and (3) a 10% probability of remaining unchanged. In the sentence-level representation, the Next Sentence Prediction method is devoted to training. Specifically, in the sentence-level prediction task, that is, to predict whether two sentences are a continuous sentence-pair relationship. When choosing sentences A and B for each pre-training example, there is a 50% probability B is the actual next sentence that follows A, and 50% of the probability is a random sentence from the corpus. The above two training tasks could respectively capture the representations of the word-level and the sentence-level and can be jointly trained. The structure of the BERT pre-training language model is shown in Figure 1.

The characteristics of the BERT pre-training language model have been introduced above. For a given token, the input layer representation is constructed by adding token embeddings, segment embeddings, and position embeddings. The advantage of this model is that it could learn more comprehensive vocabulary and semantic information and resolve ambiguities. The input representation of BERT is shown in Figure 2.

Among them, the position Embedding represents the position information of each word in the sentence, while the segment Embedding represents sentence pair information. For the sentence-pair tasks, EA represents the left sentence in the sentence-pair, and EB represents the right sentence. The token Embedding represents the embedding information of the word itself.

There are many calculation methods to obtain the Position Embeddings. The calculation method adopted here is as follows:(1)$$PE\left(pos,2i\right)=\sin\left(pos/10,000^{2i/d_{model}}\right)$$
(2)$$PE\left(pos,2i+1\right)=\cos\left(pos/10,000^{2i/d_{model}}\right)$$
where *pos* is the position of the word in the sentence, and *i* is the dimension of the vector.

### 3.2. Slice Recurrent Neural Network

RNN has been widely used in NLP tasks because of its ability to capture the temporal characteristics of the sequences, but it has the disadvantage of slow training speed. In order to solve the shortcomings of RNN, Yu et al. [9] proposed the SRNN structure to make parallel training possible. By cutting the input sequence, the recurrent units can simultaneously process the sub-sequences on each layer, which greatly improves the training speed of the model. At the same time, SRNN can capture the complete information of the sequence. Its structure is shown in Figure 3.

SRNN cuts the original sequence into multiple minimum subsequences of the same length. The characteristic information extracted from each layer can be transmitted in the network.

Given an input sequence, X of length T is expressed as:
(3)$$X=[x_{1},\;x_{2},\;\ldots ,\;x_{T}]$$
where *x* represents the input at each time step.

Then the input sequence X is divided into *n* equal-length subsequences, and *N* is used to represent the length of a single subsequence, which is:(4)$$N=\frac{T}{n}$$
where *n* is the number of slices, so the sequence X can be expressed as:(5)$$X=\left[N_{1},N_{2},\ldots ,N_{n}\right]$$

Next, each subsequence *N* is divided into *n* equal-length subsequences again. The slicing operation is repeated *k* times until it is cut into the required minimum subsequence. The length of the minimum subsequence can be set as:(6)$$l_{0}=\frac{T}{n^{k}}$$$s_{0}$ is the number of minimum subsequences, which is:(7)$$s_{0}=n^{k}$$

The above is an introduction to the principle of SRNN. As shown in Figure 3, the length of the input sequence *T* is 8, and the number of slices in each layer *n* is 2. After two slicing operations, the original input sequence is cut into four minimum sub-sequences, and the length of each sub-sequence is 2.

### 3.3. Multi-Head Self-Attention Mechanism

The attention mechanism plays an important role in the field of NLP [31]. From the initial application of the alignment model in machine translation [32] to the MSA mechanism adopted by the BERT model recently suggested by Google, its application scope is becoming more and more popular. Firstly, we introduce the Self-attention mechanism.

The Self-attention mechanism, known as the internal attention mechanism, was proposed by Google in 2017. In order to better understand the Self-attention mechanism, the schematic diagram is shown in Figure 4. The attention mechanism is essential to calculate the weight of a series of Key-Value pairs in a Query. In the traditional attention mechanism, the Key represents the source element. Value represents the destination element. The output is computed as a weighted sum of the values, where the weight assigned to each value is computed by a compatibility function of the query with the corresponding key. While in the Self-Attention mechanism, the elements represented by Query, Key, and Value are consistent. This method could obtain the relationship between the elements within the sentence and thus catch the internal structure of the sentence.

In the Self-attention mechanism, the similarity function utilizes the scaling dot product.
(8)$$Attention\left(Q,K,V\right)=softmax\left(\frac{QK^{T}}{\sqrt{d_{k}}}\right)V$$
where *Q*, *K*, and *V* denote the query, key, and value matrix, correspondingly. And $d_{k}$ is the dimension of *K*.

The MSA mechanism is actually based on the Self-attention mechanism to perform multiple calculations. It allows the model to jointly process information from various representation subsequences at various locations. Firstly, the Query, the Key, and the Value have transformed through *H* times linear transformation. Here, *H* is the abbreviation of Head. And *H* is the number of linear transformations. Then, the linear transformation value is used as the input of the dot-product to calculate the attention weight of H times. Finally, stitch the attention weights calculated after H times, and then the linear transformation is performed again, and the linear transformation value is used as the weight of the MSA mechanism. The Multi-Head is calculated as the following formula:(9)$$MultiHead\left(Q,K,V\right)=Concat\left(head_{1},\ldots ,head_{h}\right)W^{O}$$
(10)$$head_{i}=Attention\left(QW_{i}^{Q},KW_{i}^{K},VW_{i}^{V}\right)$$

Among them, $W_{i}^{Q}$,$W_{i}^{K}$,$W_{i}^{V}$ and $W^{O}$ are parameter matrices. The structure of the Multi-head Self-attention mechanism is shown in Figure 5.

### 3.4. Multi-Head Self-Attention Aware Bi-SGRU Classification Model

In the current research on sentiment classification, the mainstream sentiment classification models adopt a combination of bidirectional RNN and attention mechanism. In order to further improve the classification performance, we propose a Bi-SGRU sentiment classification model integrated with the MSA mechanism and BERT embedding. For the multi-head attention mechanism, it can establish multiple sub-spaces to make the model pay attention to various aspects of text information. If the length of the input sequence, *n*, is relatively large, instead of calculating attention with all of the words, each word is calculated with restricted *r* words, respectively. The value of *r* is much smaller than n. That means, we can say, with the utility of the MSA, we could handle the MSA of a long sequence with much shorter sequences. In terms of parallelism, multi-head attention, such as CNN, does not rely on the previous calculations and can be parallelized very well, which is better than the RNN-style network. At the same time, multi-head attention is better than CNN, which can only learn local information. A gated recurrent neural network is an improved recurrent neural network structure, which solves the problem of large time step dependence in time series, and the bidirectional gated recurrent neural network can better capture the dependence between words relationship. As a result, the bidirectional slice-gated recurrent neural network can not only greatly shortens the time but also has the advantage of better capturing the dependencies between the words. The structural framework is shown in Figure 6.

It can be seen from Figure 6 that the original sequence *X* is used as the input information of the model input layer, which can be expressed as [$x_{1},x_{2},\ldots x_{T}$].

Firstly, let the original sequence *X* pass through the embedding layer. Each word in the original sequence *X* needs to be converted into the representation of the BERT word vector, and this sequence is applied as the input information of the bidirectional SGRU network. Secondly, the SGRU network is sliced into subsequences of equal length. The sequence length *T* is 8, and the length of the smallest subsequence is 2. Then $x_{i}$ is the *i*-th text and $x_{ij}$ is the word vector of the *j*-th word in the *i*-th text. Through the slicing method, the training speed can be greatly accelerated. The hidden state of the subsequences on 1th layer of the bidirectional SGRU network is:(11)$$\vec{h_{ij}^{1}}=\vec{GRU}\left(x_{ij},\vec{h_{i\left(j-1\right)}^{1}}\right)$$
(12)$$\overset{\leftarrow }{h_{ij}^{1}}=\overset{\leftarrow }{GRU}\left(x_{ij},\overset{\leftarrow }{h_{i\left(j-1\right)}^{1}}\right)$$

Taking the state of the first hidden layer of the bidirectional SGRU network as the input of the MSA mechanism layer, the output state obtained through the MSA mechanism is as follows:(13)S=MultiHead(H,H,H)
where *H* is the hidden layer state of each smallest subsequence of the bidirectional SGRU network.

The output sequence *S* is obtained by the MSA mechanism. And it is the input value of the next layer of the bidirectional SGRU network to obtain the hidden layer state. The calculation formula is as follows:(14)$$\vec{h_{t}^{2}}=\vec{GRU}\left(s_{t\times \frac{m_{0}}{m_{1}}-l_{0}+1}\sim s_{t\times \frac{m_{0}}{m_{1}}}\right)$$
(15)$$\overset{\leftarrow }{h_{t}^{2}}=\overset{\leftarrow }{GRU}\left(s_{t\times \frac{m_{0}}{m_{1}}-l_{0}+1}\sim s_{t\times \frac{m_{0}}{m_{1}}}\right)$$
where $h_{t}^{2}$ is the hidden layer state of the *t-th* subsequence in the second layer of the SGRU network, $m_{0}$ is the number of minimum subsequences, $m_{1}$ is the number of subsequences in the next layer, and $l_{0}$ is the length of the minimum subsequence.

Through analogy analysis, the other hidden layer states in the SGRU network are as follows:(16)$$\vec{h_{t}^{n}}=\vec{GRU}\left({\vec{h}}_{t\times \frac{m_{n-1}}{m_{n}}-l_{n}+1}\sim {\vec{h}}_{t\times \frac{m_{n-1}}{m_{n}}}\right)$$
(17)$$\overset{\leftarrow }{h_{t}^{n}}=\overset{\leftarrow }{GRU}\left({\overset{\leftarrow }{h}}_{t\times \frac{m_{n-1}}{m_{n}}-l_{n}+1}\sim {\overset{\leftarrow }{h}}_{t\times \frac{m_{n-1}}{m_{n}}}\right)$$

The final output of the model is obtained by Equations (16) and (17). The output of the forward network is connected with the backward network, and a softmax layer is added after the final splicing layer to classify the labels. The calculation formula for this part is as follows:(18)$$Y=concat\left(\vec{Y},\overset{\leftarrow }{Y}\right)$$
(19)$$P=softmax\left(W_{Y}Y+b_{Y}\right)$$
where $W_{Y}$ and $b_{Y}$ are the training parameters of the model.

## 4. Experiment Analysis

### 4.1. Dataset

In the experiments, two comment datasets about public opinion are utilized, namely the Yelp 2015 dataset and the Amazon dataset. The Yelp 2015 dataset, a subset of the Yelp Challenge, contains 1569 k pieces of review data, while the Amazon dataset has 3650 k pieces of product review data. The two datasets are divided into three groups by 8:1:1, in which 80% of the data are used as the training set, 10% as the validation set, and 10% as the test set. The relevant information from the two datasets is shown in Table 1.

### 4.2. Experimental Environment

This experiment adopts Python language for programming. The deep learning framework applied in the experiment is Keras, based on Tensorflow. The entire experiment was compiled with PyCharm. The computer configuration employed in the experiment is shown in Table 2.

### 4.3. Evaluation Metric

The evaluation indicators adopted in the experiment include classification accuracy, training loss value, and training time. Accuracy is the most intuitive and important evaluation method in classification tasks. The performance of the model can be directly reflected by the classification accuracy. The calculation formula is as follows:(20)$$\mathrm{Accuracy}=\frac{N_{T}}{N_{M}}$$
where $N_{T}$ is the number of correctly classified samples, and $N_{M}$ represents the total number of samples. The training loss value is the difference between the predicted value of the model and the true value. The smaller the loss is, the better the convergence of the model is. The calculation method of the training loss utilized in these experiments is the Cross-Entropy Loss [33]. The calculation equation is as follows:(21)$$L\left(Y,P\left(Y|X\right)\right)=-\underset{N}{\sum }YlogP\left(Y|X\right)+\left(1-Y\right)\log\left(1-P\left(Y|X\right)\right)$$

### 4.4. Parameter Configuration

The parameter configuration of the model used in the experiments is shown in Table 3.

### 4.5. Experiment Procedure

The experimental process can be roughly divided into four steps. Firstly, clean up the data to retain the required information, and the punctuation marks and other special characters are removed. Secondly, we use the BERT pre-training model to obtain the corresponding word vector representation of the text data.

Finally, we extract the semantic features of related data through the bidirectional SGRU network and utilize the MSA mechanism to better capture hidden information. Finally, the Softmax layer is applied to distinguish the emotional polarity.

To verify the effectiveness of the proposed model, we compare this model with the mainstream sentiment classification models, which are Dilated Causal Convolution Neural Network (DCCNN) proposed by Oord et al. [34], LSTM [35], GRU [36], SGRU [37], and Bi-LSTM-Attention [38].

Traditional RNNs perform poorly in solving the association between long sequences. The reason is that when backpropagating, the calculation of the gradient of the long sequence is very likely to be abnormal such as the gradient disappearing or exploding. The total amount of model parameters of LSTM or GRU is relatively small. Hence the performance and effect are excellent on short-sequence tasks. However, due to the relatively complex internal structure of LSTM, the training efficiency is much lower than that of traditional RNNs under the same computing power. GRU cannot completely solve the problem of gradient disappearance. One of thedrawbacks of RNN is that it cannot be calculated in parallel. The BiLSTM-Attention and the proposed Sliced Bi-GRU-Attention are both bidirectional, and the dependency between captured words is improved. But the problems with LSTM and GRU still cannot be handled. Our Sliced Bi-GRU is currently the most advanced and efficient solution for the analysis of long text, but it also has the problem of not being able to fully extract the semantic features of the text.

### 4.6. Experimental Results

In order to verify whether the proposed model can converge during the training process, experiments were conducted utilizing the Yelp 2015 comment dataset. Through experiments, the model performed 10 iterations during the training process and recorded the loss value. It is compared with four baseline models of LSTM, GRU, SGRU, and Bi-SGRU. The comparison of the loss curve is shown in Figure 7. Here, the slicing method adopted by the slicing neural network is SGRU (8,2). The slice operation time *k* is 2, and the slice number *n* is 8.

It can be seen from Figure 7 that although the convergence speed of this model is slower than other models, as the number of iterations increases, the loss value of this model is lower than the other four models. Among the first four iterations, the GRU benchmark model has the fastest convergence speed. The model applied in this article reached the lowest loss value in the eighth iteration, and the loss value was still the lowest in subsequent iterations. The reason for the slow convergence of the model is that the trainable parameters of the model are larger than those of other models. Therefore, after a certain number of iterations, this model will show its advantages.

In the sentiment classification task, to further verify the effectiveness of the model, the model was compared with some mainstream models. The classification accuracy and the time required for one iteration are applied as evaluation conditions. Compared with the method proposed in [39], the method proposed in this paper uses a slicing network method. It can be seen from Table 4 that the training time has been greatly improved. The experimental results are shown in Table 4.

It can be seen from Table 4 and Table 5 that, compared with the Bi-GRU model proposed by Sachin S et al. [40], although our model is slightly weaker in accuracy, it has greatly improved the training speed. It can be seen from Table 5 that in the Yelp 2015 dataset and the Amazon dataset, the model used in this article is superior to other models in classification accuracy, with accuracy rates of 74.37% and 62.57%, respectively. Among all the models, the DCCNN model has the lowest classification accuracy. This is because DCCNN is unable to capture the timing information of the sequence, resulting in relatively few effective features extracted by the model, so the classification performance of the model is poor. Compared with DCCNN, LSTM and GRU have better classification accuracy. The SGRU model has also obtained high classification accuracy, so the feature information can be captured by slicing the sequence. At the same time, the feature information is transmitted through multiple network layers, so the upper-layer network can obtain more useful information. The Bidirectional LSTM model with an attention mechanism also obtained higher classification accuracy. It can be seen that the attention mechanism could gain more useful information and improve the performance of the model.

It can be seen from the experimental results that although the iteration speed of the DCCNN model is the fastest. But the accuracy rate of the DCCNN model is the lowest. Compared with the DCCNN model, the sliced neural network can greatly improve the accuracy rate under the premise of ensuring training speed. From the experimental results, the training speed of the SGRU model is 8–13 times faster than the GRU model and 9–16 times faster than the LSTM model. Compared with the STSM and other models proposed by Yang et al. [41], the training speed is also improved. Although the training speed of the model is not the fastest, considering the two indicators of classification accuracy and training speed, the model reflects the best performance.

This article verifies the validity of the model from the loss value, classification accuracy, and iteration speed. In order to consider the impact of more factors on the performance of the model, we will start with different slicing methods and verify the performance differences of models under different slicing methods through comparative experiments. Experiments were conducted on the Yelp 2015 dataset and the Amazon dataset. The results on each dataset are shown in Table 5.

It can be seen from Table 5 that SGRU (16,1) gets the highest accuracy on the Yelp 2015 dataset and the Amazon dataset, while SGRU (4,3) takes the shortest time to iterate on these two datasets. A model with a faster training speed cannot guarantee the best classification accuracy. Therefore, for different tasks, different slicing methods can be selected to meet different actual needs.

The number of heads of the MSA mechanism is also an important parameter that affects the performance of the model. In order to explore the influence of different Attention heads on classification, this article conducts experiments on the Yelp 2015 dataset and utilizes the model in this experiment with the different values of Attention mechanism heads (2,4,8,12,16). The experimental results are shown in Figure 8.

It can be seen from the results that compared with other models, the head of 12 shows better performance. It is also a manifestation of the MSA mechanism. With more heads, the MSA mechanism can learn more different subspace information. But it does not mean the performance will obtain better with more Attention mechanism heads. The training parameters also increase. Therefore, for different datasets, only through multiple comparison experiments can we obtain the appropriate number of heads and the best experimental results.

## 5. Conclusions

This article proposes a Sliced Bi-GRU model that combines the Multi-head Self-attention mechanism and BERT embedding. Compared with mainstream sentiment classification models, it shows a higher classification accuracy. It solves the shortcomings that RNN cannot be parallelized during the training process and greatly improve the training speed of the model. In general, the word vector representation in the BERT pre-training language model proposed in this paper can better reflect the semantic information of the text. The model also changes the traditional RNN connection method, accelerates the training speed through slicing operations, and ensures better classification accuracy. However, with further in-depth research on the preprocessing Embedding technologies such as BERT word vector model, the [MASK] marking method introduced in BERT has inconsistent usage patterns in two stages, which may bring some performance loss. Therefore, we will further build a better classification model to overcome such kind of limitations.

## Figures and Tables

**Figure 1 sensors-23-01481-f001:**
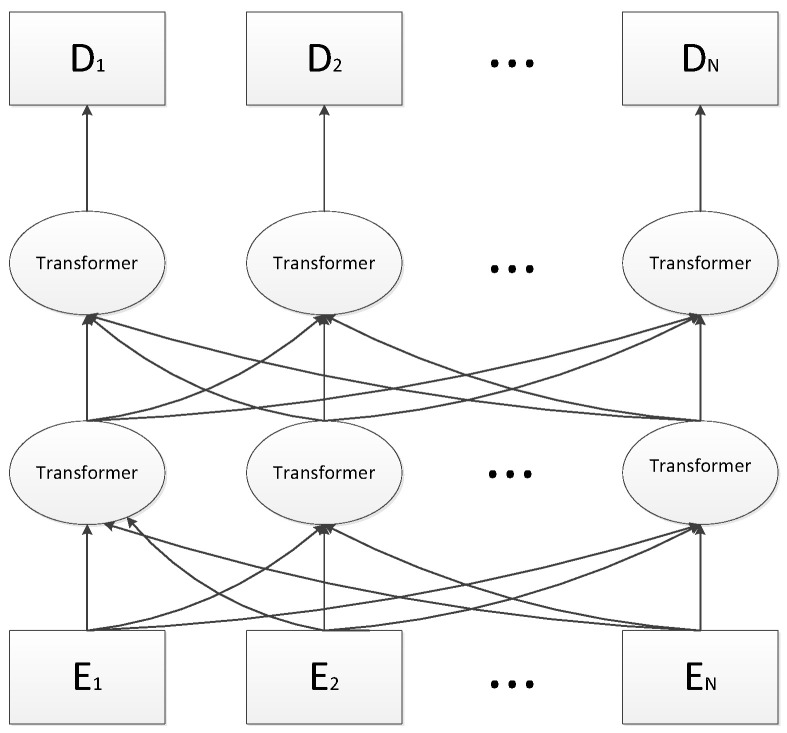
The BERT pre-trained language model [16].

**Figure 2 sensors-23-01481-f002:**
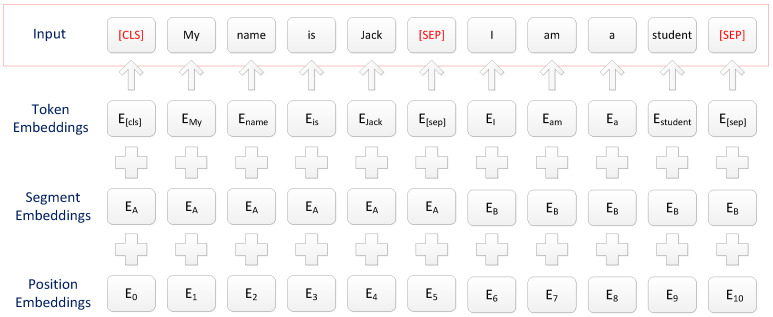
BERT input representation [16].

**Figure 3 sensors-23-01481-f003:**
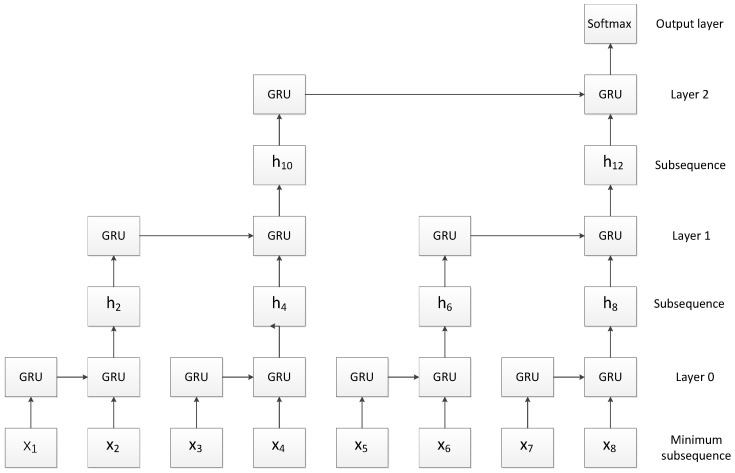
SRNN structure [9].

**Figure 4 sensors-23-01481-f004:**
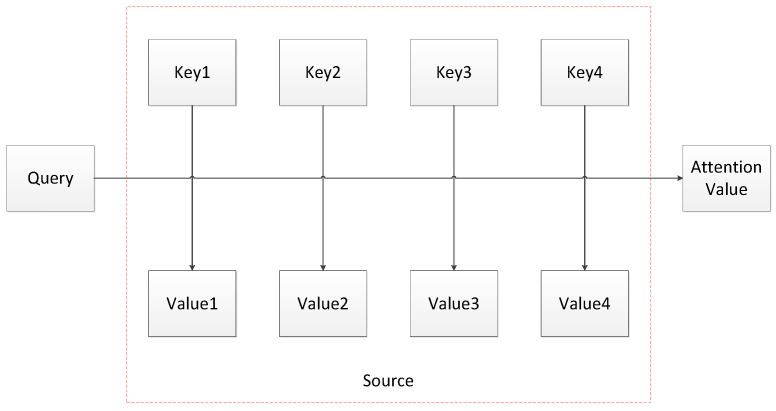
The Structure diagram of the attention mechanism.

**Figure 5 sensors-23-01481-f005:**
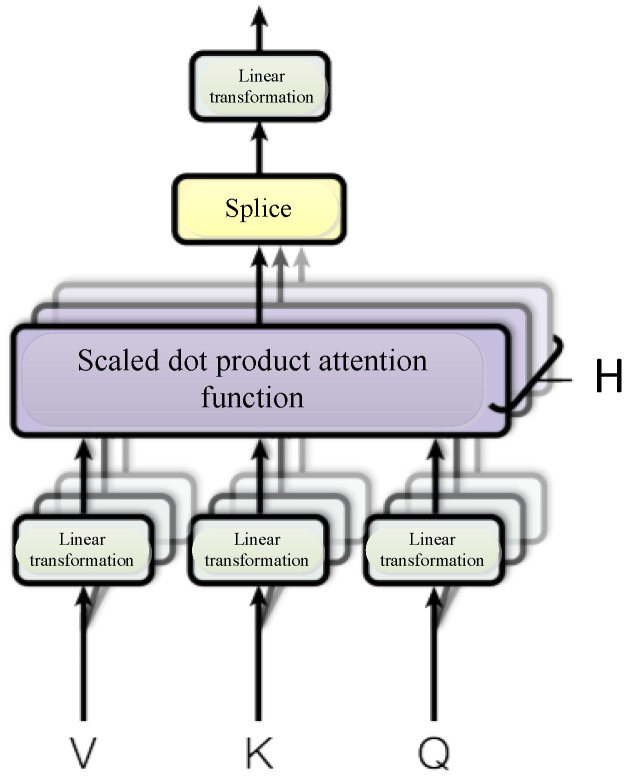
Multi-head Self-Attention mechanism.

**Figure 6 sensors-23-01481-f006:**
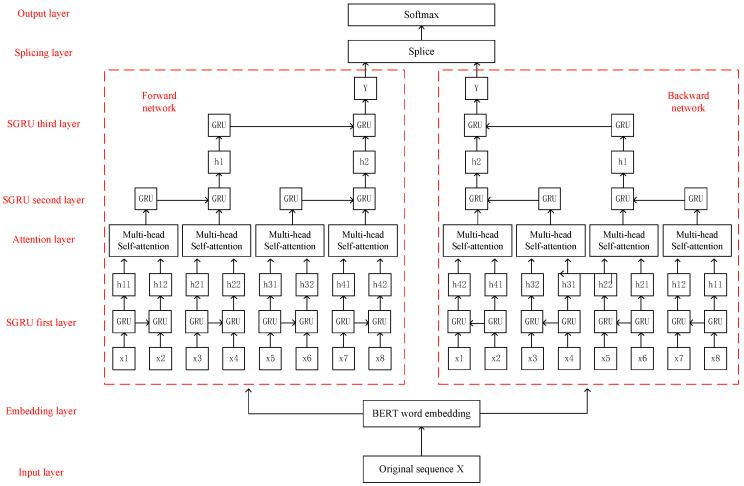
Sliced Bi-GRU classification model based on the Multi-head Self-Attention mechanism.

**Figure 7 sensors-23-01481-f007:**
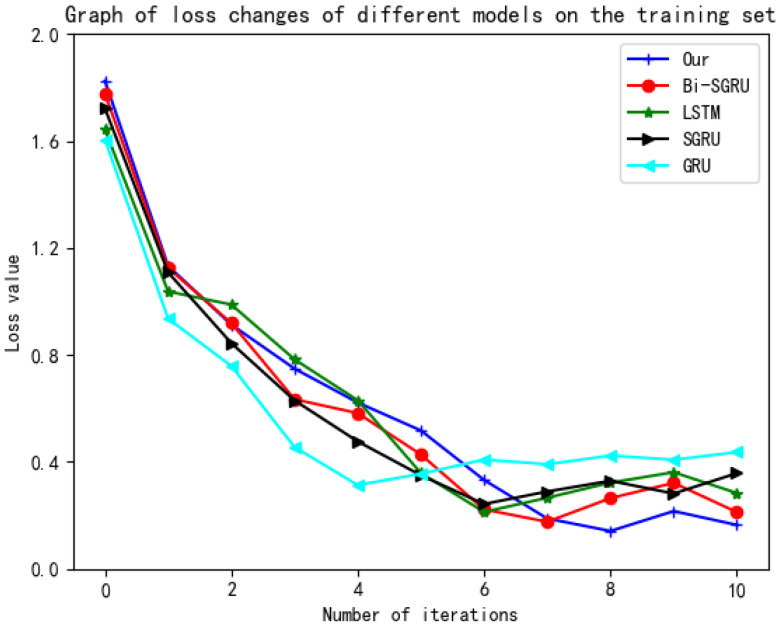
Loss curves of different models on the training set.

**Figure 8 sensors-23-01481-f008:**
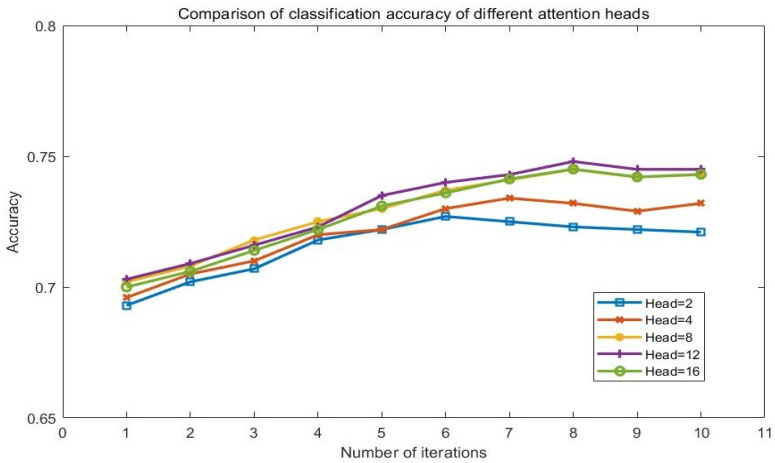
Comparison of accuracy of different attention heads.

**Table 1 sensors-23-01481-t001:** Relevant information about the experimental dataset.

Dataset Name	Max Words	Average Words	Vocabulary	Category
Yelp 2015 dataset	1092	108	228,715	5
Amazon dataset	441	83	1,274,916	5

**Table 2 sensors-23-01481-t002:** Experimental platform configuration.

Hardware Information	Related Configuration
Operating system	Windows 10
CPU	Intel i5-7200
RAM	8G
Graphics card	NVIDIA GeForce GTX1070
Graphics card acceleration	CUDA 10.1/cudnn 7.1

**Table 3 sensors-23-01481-t003:** Hyperparameter Setting.

Parameter Name	Parameter Value
Word vector dimension	768
Batch_size	100
Number of filters	50
Number of Attention head	12
Number of iterations	10
Optimizer	Adam
Dropout rate	0.5
Learning rate	0.001
Loss function	Cross-Entropy Loss

**Table 4 sensors-23-01481-t004:** Performance comparison of the models on two public datasets. Results that surpass all competing methods are **bold**.

Dataset	Model Name	Accuracy (%)	The Time Required for One Iteration (min)
Yelp 2015 dataset	DCCNN [34]	71.83	4.25
	LSTM [35]	74.05	163.38
	GRU [36]	73.76	136.74
	SGRU [37]	73.91	10.33
	BiLSTM-Attention [38]	74.18	277.75
	BiLSTM with Attention [39]	74.22	219.84
	Our	**74.37**	25.79
Amazon dataset	DCCNN [34]	60.18	14.70
	LSTM [35]	62.12	385.11
	GRU [36]	61.97	322.25
	SGRU [37]	62.03	40.05
	BiLSTM-Attention [38]	62.29	638.09
	BiLSTM with Attention [39]	62.34	77.70
	Our	**62.57**	101.83

**Table 5 sensors-23-01481-t005:** The accuracy and training time of the models on each dataset. Results that surpass all competing methods are **bold**.

Dataset	Model	Accuracy (%)	Time (min)
Yelp 2015 dataset	SGRU(2,6)	73.92	31.25
	SGRU(4,3)	74.33	**24.98**
	SGRU(8,2)	74.37	25.79
	SGRU(16,1)	**74.55**	39.23
Amazon dataset	SGRU(2,6)	62.05	118.32
	SGRU(4,3)	62.54	103.52
	SGRU(8,2)	62.57	**101.83**
	SGRU(16,1)	**62.80**	137.63

## Data Availability

The datasets used during the current study are the Yelp 2015 dataset and the Amazon dataset, which are available online at https://www.yelp.com/dataset (accessed on 20 November 2022), and https://snap.stanford.edu/data/web-Amazon.html (accessed on 20 November 2022), respectively.

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
