# Peer review of "Text Sentiment Classification Based on BERT Embedding and Sliced Multi-Head Self-Attention Bi-GRU"

_sensors, 2023, doi:10.3390/s23031481_

Round 1

Reviewer 1 Report

The work is very interesting, well structured and follows a logical sequence. 

The methodology is correct and the results are clearly presented. 

The work lacks to add in results the real improvement that this method supposes, that is to say, to indicate clearly how much your method improves in an example close to reality, in such a way that the reader can get a clear idea of the improvement obtained. 

Author Response

Point 1: The work is very interesting, well structured and follows a logical sequence. The methodology is correct and the results are clearly presented. 

 Response 1: Thank you for your valuable comments.

Point 2: The work lacks to add in results the real improvement that this method supposes, that is to say, to indicate clearly how much your method improves in an example close to reality, in such a way that the reader can get a clear idea of the improvement obtained. 

Response 2: We appreciate your professional comments on this article. Your comments are very helpful to improve this article.

The reviewer suggested adding the actual improvement of the proposed method in an example close to reality in the article. Thank you again for your insightful suggestions. That is a very good advice.

Our model achieves better results than the baseline model on Yelp2015 and Amazon datasets. The Yelp review data dataset is accurate review data collected from the Yelp website, comprising approximately 160,000 merchants, 8.63 million reviews, and 200,000 picture data from eight metropolitan areas. It is mainly the user's score on hotels or merchants. The Amazon review data set is compiled and published by Mcauley on Amazon's website based on accurate comments. These datasets can be considered to be exactly the real instances. As a result, they are widely used in text classification tasks, such as Z.C. Yang et al. [23], Zhang et al. [17] and Yu et al. [9]. The obtained improvements of the proposed algorithm can be seen from the performance comparison tables.

Reviewer 2 Report

In this work, Sliced Multi-head Self-attention Bi-GRU and BERT Embedding were utilized to classify sentiment more quickly and accurately. In order to improve the likelihood of parallel processing, sentences were seperated and learnt simultaneously in adding to using BERT embedding to tackle the unknown word problem in word embedding. The repective mehodologies were effectively merged into one model in order to handle connected problems collectively. However, authors would like to answer a few questions below.

1. Can you claim that the proposed approach in Figure 7 has undergone the best optimization? The suggested approach, as noted in the study, takes the longest to optimize and the loss is not noticeable less that that of Bi-SGRU or SGRU.

2. The accuracy of the two datasets in Table 4 has a correlation coefficient that is very near to 1. This section needs to be explained.

3. The Amazon dataset does not demonstrate any benefits of the proposed study in comparison to the Yelp dataset. More research is necessary for this section. And just using GRU makes more sense, even with Yelp datasets.

Author Response

In this work, Sliced Multi-head Self-attention Bi-GRU and BERT Embedding were utilized to classify sentiment more quickly and accurately. In order to improve the likelihood of parallel processing, sentences were seperated and learnt simultaneously in adding to using BERT embedding to tackle the unknown word problem in word embedding. The repective mehodologies were effectively merged into one model in order to handle connected problems collectively. However, authors would like to answer a few questions below.

Point 1: Can you claim that the proposed approach in Figure 7 has undergone the best optimization? The suggested approach, as noted in the study, takes the longest to optimize and the loss is not noticeable less that that of Bi-SGRU or SGRU.

Response 1: We appreciate your professional comments on this article. Your comments are critical to our articles. All these comments have made outstanding contributions to improving the quality of our articles. After this revision, we have written a point-by-point reply letter, as you can see.

First, our model mainly comprises BERT Embedding, Bi SGRU unit, and MSA. BERT word vector is used as the pre-training input of the model, and then the sliced Bi-SGRU and MSA mechanism are used for text emotion classification. The sliced Bi-GRU unit combines the advantages of slicing operation, bidirectional GRU. To verify the impact of different SGRU slicing methods on model accuracy and training time, we conducted experiments on Yelp 2015 data and Amazon data sets. Based on the model accuracy and training time conditions, we conclude that the optimal slicing method is the sliced operation time k is 2 and the slice number n is 8, as shown in Table 5. Then, to verify the influence of different head numbers of MSA on the model. We use different heads of  MSA to conduct experiments on the dataset. The experiment shows that the number of heads of MSA 12 is the optimal parameter, as shown in Figure 8. Finally, the Adma function is used to optimize the other experimental parameters. In addition, as shown in Figure 7, since the number of trainable parameters of our model is much larger than those of other models, the initial convergence speed is truly slower than that of Bi SGRU and SGRU models. However, as the number of iterations increases, the loss value of the proposed model is always lower than other models. In particular, our model reached the lowest loss value of 0.12 at the eighth iteration, which is only half of the loss value of Bi SGRU.

The description in the original manuscript is not very clear. We have modified the description in the subsection 4.6 of the original text.

Point 2: The accuracy of the two datasets in Table 4 has a correlation coefficient that is very near to 1. This section needs to be explained.

Response 2: Thank you again for your valuable comments to improve the quality of our manuscript. As the reviewer pointed out, the correlation coefficient of the accuracy of the two data sets in Table 4 is close to 1. In my opion, because Yelp review data and Amazon review data have the nearly the same data presentation form, both of which are composed of goods and reviews, the performance of different baseline models on the two datasets is similar. In fact, the similar data distribution plays the important role. In addition, as can be seen in other recently published articles, the correlation coefficient of the accuracy of different models on these two data sets also approximate to 1, for example, Z.C. Yang et al. [23] Hierarchical attention networks for document classification, Zhang et al. [17] Character level Collaborative Networks for Text Classification and Yu et al. [9] Sliced Recurrent Neural Networks.

Point 3: The Amazon dataset does not demonstrate any benefits of the proposed study in comparison to the Yelp dataset. More research is necessary for this section. And just using GRU makes more sense, even with Yelp datasets.

Response 3: Thank you very much for your professional advice. Maybe Our descriptions of the experimental results are not very clear so that the reviewer misunderstand it. We have revised the descriptions of the experimental analysis. The proposed method achieves the highest accuracy on the Amazon dataset compared with other baseline models. Compared with the GRU baseline model and SGRU baseline model, our model has improved the accuracy by 0.6% and 0.5%, respectively. The training time is only one-third of that of the GRU baseline model. Although the training time is not the shortest among all the methods, our method achieves the best effect by integrating model accuracy and training time indicators. In the Yelp dataset, only the GRU baseline model was used to achieve an accuracy of 73.76%, but the training time was up to 136.74 minutes. However, our model improved the accuracy rate by 0.5% in the Yelp dataset, at the same time reduced the training time to one-sixth of the original, as shown in the Table 4.

Round 2

Reviewer 2 Report

The authors have revised the paper by fully considering the referee's points. Also, the referee's doubts were well explained. Therefore, there is no objection to the publication of this paper.